# Synthesis, Physicochemical Characterization, Biological Evaluation, In Silico and Molecular Docking Studies of Pd(II) Complexes with P, S-Donor Ligands

**DOI:** 10.3390/ph16060806

**Published:** 2023-05-29

**Authors:** Hizbullah Khan, Muhammad Sirajuddin, Amin Badshah, Sajjad Ahmad, Muhammad Bilal, Syed Muhammad Salman, Ian S. Butler, Tanveer A. Wani, Seema Zargar

**Affiliations:** 1Department of Chemistry, University of Science and Technology, Bannu 28100, Pakistan; 2Department of Chemistry, Quaid-i-Azam University, Islamabad 45320, Pakistan; aminbadshah@qau.edu.pk; 3Department of Health and Biological Sciences, Abasyn University, Peshawar 25000, Pakistan; sajjademman8@gmail.com; 4Department of Chemistry, Kohat University of Science and Technology, Kohat 26000, Pakistan; mbilal@kust.edu.pk; 5Department of Chemistry, Islamia College University, Peshawar 25120, Pakistan; salman@icp.edu.pk; 6Department of Chemistry, University of McGill, Montreal, QC H3A 0B8, Canada; ian.butler@mcgill.ca; 7Department of Pharmaceutical Chemistry, College of Pharmacy, King Saud University, P.O. Box 2457, Riyadh 11451, Saudi Arabia; twani@ksu.edu.sa; 8Department of Biochemistry, College of Science, King Saud University, P.O. Box 22452, Riyadh 11451, Saudi Arabia; szargar@ksu.edu.sa

**Keywords:** Pd(II) complexes, X-ray structure, antibacterial activity, antitumor activity, in silico study, molecular docking

## Abstract

One homoleptic (**1**) and three heteroleptic (**2**–**4**) palladium(II) complexes were synthesized and characterized by various physicochemical techniques, i.e., elemental analysis, FTIR, Raman spectroscopy, ^1^H, ^13^C, and ^31^P NMR. Compound **1** was also confirmed by single crystal XRD, showing a slightly distorted square planar geometry. The antibacterial results obtained via the agar-well diffusion method for compound **1** were maximum among the screen compounds. All the compounds have shown good to significant antibacterial results against the tested bacterial strains, *Escherichia coli, Klebsiella pneumonia,* and *Staphylococcus aureus,* except **2** against *Klebsiella pneumonia*. Similarly, the molecular docking study of compound **3** has shown the best affinity with binding energy scores of −8.6569, −6.5716, and −7.6966 kcal/mol against *Escherichia coli, Klebsiella pneumonia,* and *Staphylococcus aureus*, respectively. Compound **2** has exhibited the highest activity (3.67 µM), followed by compound **3** (4.57 µM), **1** (6.94 µM), and **4** (21.7 µM) against the DU145 human prostate cancer cell line using the sulforhodamine B (SRB) method as compared to cisplatin (>200 µM). The highest docking score was obtained for compounds **2** (−7.5148 kcal/mol) and **3** (−7.0343 kcal/mol). Compound **2** shows that the Cl atom of the compound acts as a chain side acceptor for the DR5 receptor residue *Asp B218* and the pyridine ring is involved in interaction with the *Tyr A50* residue via arene-H, while Compound **3** interacts with the *Asp B218* residue via the Cl atom. The physicochemical parameters determined by the SwissADME webserver revealed that no blood-brain barrier (BBB) permeation is predicted for all four compounds, while gastrointestinal absorption is low for compound **1** and high for the rest of the compounds (**2**–**4**). As concluding remarks based on the obtained in vitro biological results, the evaluated compounds after in vivo studies might be a good choice for future antibiotics and anticancer agents.

## 1. Introduction

Cancer is a devastating disease, yet many types can be entirely treated if discovered early on, and for many others, patients’ lives can be greatly prolonged. Surgery, radiation therapy, and chemotherapy [1] are three common cancer treatment modalities that are usually used in conjunction. The form of treatment is determined by the nature of the disease and the stage at which it has progressed [2]. Chemotherapy is one of the most common treatments for numerous forms of malignancies. Localized tumors can be efficiently treated with surgery and radiotherapy; however, their success is hampered by cancer cells spreading to other parts of the body [2]. Although radiotherapy and surgery can cure 40% of cancer patients (mostly those with tiny tumors), the spread of cancer cells may result in the death of the remaining 60% [3,4,5]. Surgery and radiation therapy cannot be utilized to treat metastasized tumors; hence, chemotherapy offers this advantage [6,7]. An ideal antitumor medication would be one that destroys only malignant cells while leaving healthy cells alone.

In actuality, all anticancer medications also impact healthy cells, resulting in a variety of side effects, including nausea, vomiting, and exhaustion [2,4]. Since anticancer medicines can harm blood cells, patients may develop anemia and a reduced white blood cell count [8]. The side effects are normally treatable and diminish once the treatment is completed. Combination chemotherapy combines medications with diverse methods to overcome resistance, reduce adverse effects, and kill as many malignant cells as possible [9]. Therapeutic medications cause DNA damage either directly or indirectly by interfering with biomolecules that are utilized to manufacture DNA, such as enzymes and proteins [10].

Many efforts have been made to clarify cisplatin’s biological targets since its discovery as an anticancer medication [11]. Cisplatin has the potential to target DNA, RNA, and glutathione, although DNA is likely to be the most significant target [12]. We now have a fairly thorough grasp of how cisplatin and its analogues interact with DNA and what structural alterations take place as a result of this interaction after decades of research [13,14]. Although only around 1% of cisplatin molecules attach to DNA [15], the anticancer action of cisplatin is thought to be predominantly due to DNA binding [16]. With the N7 of guanine and adenine, Pt^2+^ produces both nonfunctional and bifunctional adducts.

Khan et al. [17,18,19] prepared palladium(II) complexes using three antibiotics: tetracycline, doxycycline, and chlortetracycline. Elemental studies, conductivity analyses, thermogravimetry, and infrared spectroscopy were used to properly describe the complexes. ^1^H NMR was used to explore the mechanism of interaction between Pd(II) ions and tetracycline. Through the amide and hydroxyl groups at ring A, all three antibiotics create 1:1 compounds with Pd(II). Tetracycline’s Pd(II) complex was 16 times more efficient than tetracycline at slowing down the growth of *E. coli* HB101/pBR322, a bacterial strain that is resistant to tetracycline, and it was just as successful at stifling the growth of two *E. coli* susceptible bacterial strains. In the resistant strain, the palladium(II) combination with doxycycline increased its activity by a factor of two [20]. At concentrations of (4.6–9.1) × 10^−4^ mol/L, several palladium (II) compounds showed broad-spectrum antibacterial action against a variety of human diseases [21,22]. Dithiocrabamates are highly versatile ligands that form stable complexes with transition elements and with the majority of the main group elements, lanthanides and actinides [23,24]. The orgaotin(IV) dithiocarbamate complexes have good coordination chemistry, stability, and diverse molecular structures, which make them suitable for diverse biological activities [25,26].

Palladium(II) and platinum(II) complexes have structural and thermodynamic similarities [27,28,29,30]. As a result, creating palladium medications with the highest level of pharmacological action is of great importance. In this investigation, several mixed-ligand palladium(II) complexes have proven to be quite promising [29,31,32]. Palladium(II) metallodrugs are suitable models for studying how biological molecules interact with one another in vivo due to their greater liability (103 times quicker on average than platinum) and ease of hydration in vitro [33]. It is hypothesized that palladium complexes are more successful in treating cisplatin-resistant gastrointestinal malignancies [17,34]. Dithiocarbamates have shown medicinal importance in various fields such as agriculture, pharmaceuticals, and medicine, especially in anticancer, antioxidant, and antimicrobial activities. So keeping in view their biological importance, we have selected dithiocarbamate-based ligands to synthesize Pd(II) complexes with phosphorus donor ligands.

## 2. Results and Discussion

The physical data of the reported compounds, including molecular formula, yield, melting point, and elemental analysis, is given in the Section 3. The sharp melting points and close matching of the elemental analysis data confirm the purity of the prepared compounds. The compounds were soluble in common organic solvents and stable at room temperature.

### 2.1. FT-IR and Raman Results

The FT-IR and Raman data of the prepared compounds are given in the Section 3. A peak in the range of 2852–2935 cm^−1^ and 3040–3052 cm^−1^ was attributed to the sp^3^ hybridized aliphatic C-H and the sp^2^ hybridized aromatic C-H, respectively. The Raman peaks for Pd-Cl, Pd-P, and Pd-S bonds appeared at 298–308 cm^−1^, 210–221 cm^−1^ and 380–395 cm^−1^, respectively. The appearance of the formation of the Pd-S verifies the Pd dithiocarbamate bond formation. The peak at 1092–1097 cm^−1^ was assigned to the SCS group, while that at 1520–1530 cm^−1^ was ascribed to the C
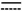
N moiety. The partial double bond character in the C-N bond for dithiocarbamate complexes was confirmed from the appearance of the peak in the range of 1520–1530 cm^−1^ as the single C-N bond appears at 1251–1361 cm^−1^ while the C=N shift value is reported at 1640–1690 cm^−1^ [17]. The presence of a single asymmetric SCS peak in the range of 1092–1097 cm^−1^ indicates the bidentate bonding mode of the dithiocarbamate ligand with palladium moiety; for monodentate bonding, two peaks are to be observed in this region if the dithiocarbamate [35]. Appendix A shows the FTIR spectrum of Complex **2**.

### 2.2. Multinuclear NMR (^1^H, ^13^C and ^31^P)

The NMR data of the ligands and their palladium complexes is given in the Section 3. The ^1^H NMR data is almost similar, with no conspicuous differentiation. The ^13^C NMR of the characteristic SCS fragment is observed in the range of 207.3–208.2 ppm and provides an important indication for the complex’s formation. A slight up field from the ligand due to the accumulation of electronic density over the carbon atom in SCS after the complexation with palladium metal was observed. The ^31^P NMR of the synthesized compounds is observed in the range of 20.3–27.5 ppm, and it also depicts a slight up field from the ligand as a result of the transfer of electron density from the ligand to the Pd^2+^ center. Appendix A show the ^1^H, ^13^C, and ^31^P NMR spectra of complex **1**.

### 2.3. Structural Study of Complex ***1***

Complex **1** single crystal XRD data was collected at 200 K using a STOE IPDS image plate detector using MoK_α_ radiation, as depicted in Table 1. The STOE X-AREA application was used to get cell parameters, gather data, and integrate data. With hydrogen atoms in optimal locations, all non-H atoms were refined anisotropically after the structure was solved and refined using SHELXT [36] and SHELXL-2015 [37].

The ORTEP view of complex **1** in a monoclynic crystla system with the P21 space group is given in Figure 1. The dimensions of the unit cell are: a = 16.3128(4)Å, b = 10.2144(2)Å, and c = 18.7283(4)Å, while the α and γ = 90° and β = 115.793(1)°. The crystal unit contains two independent molecules. The environment around the palladium atom is square planar with slight distortion, having bond angles of Cl1-Pd1-Cl2 = 178.40° and P(2)-Pd1-P(1) = 179.39°, as given in Table 2. Similarly, the geometry around the P1 atom is distorted tetrahedral, with bond angles of C(21)-P(1)-C(31) = 102.4°, C(21)-P(1)-C(11) = 106.8°, and C(21)-P(1)-Pd1 = 117.5°. The geometrical arrangement around the P2 atom is also distorted tetrahedral, with the largest distortion being caused by the palladium atom due to its large size, and the bond angles are: C(51)-P(2)-Pd1 = 118.7°, C(51)-P(2)-C(61) = 101.0°, and C(51)-P(2)-C(41) = 105.9°. The hydrogen bonding and other short contacts in a molecule affect the phyiscochemical properties of the molecule. The synthesized compound of palladium shows no hydrogen bonding, but it exhibits short contacts around some outer hydrogen and carbon atoms present in the propyl and benzene rings. The torsion angle for the atoms Cl1-Pd1-P(1)-C(21) is −30.7(5)°, which is synclinal; Cl2-Pd1-P(1)-C(21) is 147.7(5)°, which is anticlinal; and P(2)-Pd1-P(1)-C(21) is −74(11)°, which is also anticlinal.

### 2.4. Biological Investigation Results

#### 2.4.1. Antitumor Results

Table 3 describes the anticancer efficacy of the Pd(II) complexes against cisplatin-resistant DU145 human prostate carcinoma cells. The screened compounds have shown activity in the order of **2** > **3** > **4** > **1**. The higher activity of the heteroleptic compounds **2**, **3**, and **4** containing different organophosphine ligands despite sharing a dithiocarbamate group is due to the electronegative nitrogen atom included in the organophosphine ligand, which can interact with the DNA strands of the carcinoma cells in a variety of ways’. This may be responsible for the greater activity of compound **2**.

#### 2.4.2. Antibacterial Results

Table 4 describes the antibacterial potentials of the Pd(II) complexes against various Gram-positive and negative bacterial strains. Compound **1** is the most active, while complex **3** is the least active compound. The higher activity of complex **1** may be attributed to the fact that it has no dithiocarbamate ligand, while the other three complexes (**2–4**) have dithiocarbamate ligands, which can react with other active compounds inside the cell, so they may not have reached the target moiety.

### 2.5. In Silico Assessment

In order to make a crucial decision on a daily basis about which compound should be synthesized, tested, and promoted as a drug candidate, the study of pharmacokinetics has utmost importance. Not only the efficacy and toxicity but also other parameters such as absorption, distribution, metabolism, and excretion are thoroughly studied before the processing of a molecule to be marketed as medicine for the benefit of patients.

The pharmacokinetic parameters of the synthesized complexes were studied by *SwissADME* webserver. The physicochemical properties show that the complexes **1**, **2**, **3,** and **4** have 10, 5, 7, and 5 rotatable bonds, respectively, as shown in Table 5. The lipophilicity *LogP_o/w_* (*iLOGP*) is zero for all four complexes, while *LogP_o/w_* (*XLOGP3*) is 9.49, 6.46, 7.54, and 7.56 for complexes 1, 2, 3, and 4, respectively. Poor solubility in water is predicted, which is in agreement with the experimental observation. No blood-brain barrier (*BBB*) permeation is predicted for all four complexes, while gastrointestinal (*GI*) absorption is low for complex 1 and high for the rest of the three complexes. No drug likeness for complexes 1, 3, and 4 and yes for complex 2 is predicted by the *Lipinsky* model, while the *Veber* model has predicted positive drug likeness for all four complexes. The medicinal chemistry parameter has predicted zero alerts for all four complexes by *PAINS,* one alert for complexes 1, 2, and 4, and two alerts for complex 3 by *Brenk*.

### 2.6. Molecular Docking Results

The compounds docking results with the selected antibacterial targets are provided in Table 6. Lower binding energies indicate a stronger binding affinity of the complexes with the target proteins. Among the screened compounds, **3** have shown the highest binding affinity of −8.6569, −6.5716, and −7.6966 kcal/mol against *E. coli*, *K. pneumonia,* and *S. aureus*, respectively. The next highest binding energy is shown by compound **4** against all three bacterial strains, which is evident from the binding energy value given in Table 6. The docking confirmation and 2D interaction diagram of complex **3** against the three bacterial proteins are given in Figure 2, Figure 3 and Figure 4. Figure 2 shows that the Cl atoms of Compound 3 interact with the *His 203* and *Lys 162* residues of *E. coli*. Figure 3 shows that the benzene ring of Compound **3** interacts with the *Asn B220* residues of *K. pneumonia* via arene–H.

The interaction of the compounds **1**–**4** with DR5 (1BU3) shows that the maximum binding affinity was observed for compounds **2** (−7.5148 kcal/mol) and then for compound **3** (−7.0343 kcal/mol). The maximum in vitro antitumor activity was also observed for compound **2**. The docking results reinforced the in vitro results. The interaction with the DR5 receptor of representative compounds **2** and **3** is given in Figure 5 and Figure 6, respectively. The 2D diagram of compound **2** shows that the Cl atom of the compound acts as a chain side acceptor for the DR5 receptor residue *Asp B218* while the pyridine ring is involved in interaction with the *Tyr A50* residue via arene–H. Similarly, compound **3** interacts with the *Asp B218* residue via the Cl atom.

The interaction of the starting precursors (potassium dimethylcarbamodithioate and (PR_3_)_2_PdCl_2,_ where PR_3_ = diphenyl–2–pyridylphosphine, diphenyl–2–ethoxyphenyl phosphine, and diphenyl–*p*–tolylphosphine) was checked against the three bacteria receptors: *E. coli* (PDB_ID: 6G9S), *K. pneumonia* (PDB_ID: 4EXS), and *S. aureus* (PDB_ID: 5ZH8), as well as DR5 (1DU3).

Against *E. coli*, the best affinity was shown by the Pd(II) organophosphine precursor having a diphenyl–*p*–tolylphosphine moiety (−6.8335 kcal/mol), followed by a diphenyl-2-pyridylphosphine moiety (−6.3463 kcal/mol). Similarly, against *K. pneumonia* and *S. aureus,* the maximum binding affinity was observed for the Pd(II) organophosphine precursor having a diphenyl–2–ethoxyphenyl phosphine moiety (−5.3247, −6.1572 kcal/mol), followed by a diphenyl–*p*–tolylphosphine moiety (−5.0306, −6.0522 kcal/mol), respectively.

The best binding affinity against DR5 (1DU3) was observed for the Pd(II) organophosphine precursor having a diphenyl–2–ethoxyphenyl phosphine moiety (−7.1002 kcal/mol), followed by a diphenyl–*p*–tolylphosphine moiety (−6.9220 kcal/mol). From the comparison of binding energy, we can say that the starting Pd(II) organophosphine precursors have low binding affinity as compared to their corresponding Pd dithiocarbamate complexes. The results are shown in Appendix A.

## 3. Experimental

### 3.1. Materials and Methods

PdCl_2_ was obtained from Alfa-Aesar, whilst organophosphines, diphenyl-*n*-propylphosphine, diphenyl–2–pyridylphosphine, diphenyl–*p*–tolylphosphine, diphenyl-2–ethoxyphenyl phosphine, potassium hydroxide, hydrochloric acid, methanol, and dichloromethane were acquired from Sigma–Aldrich and put to use directly without further purification.

NMR spectra were recorded on Mercury 200 MHz and Bruker 300 MHz spectrometers. ^1^H NMR (300.13 MHz): CDCl_3_ (7.26 from SiMe4). ^13^C NMR (75.47 MHz), internal standard TMS; ^31^P NMR (121.49 MHz): CDCl_3_. IR spectra were recorded on a Nicolet 6700 FT-IR instrument in the range of 400–4000 cm^−1^ and Raman spectra (±1 cm^−1^) were measured with an InVia Renishaw spectrometer, using argon ion (514.5 nm) and near infrared diode (785 nm) lasers. Wire 2.0 software was used for the Raman data acquisition and spectra manipulations. The elemental analyses were conducted on a LECO-183 CHNS analyzer. Melting points were measured on the Stuart SMP10 apparatus and are uncorrected. GraphPad Prism was applied for statistical analysis.

### 3.2. Synthesis

The synthesized compounds were prepared in three steps. In step-1, dithiocarbamate was prepared; in step-2, palladium(II) organophosphine complexes were prepared; and in step-3, heteroleptic palladium complexes containing both organophosphine and dithiocarbamate ligands were prepared [17,18].

A carbon disulfide solution (30 mL) in dry methanol was added dropwise to dimethyl amine and potassium hydroxide dissolved in methanol (30 mL) in an equimolar ratio. The reaction mixture was stirred for 4 h at 0 °C (Figure 1). A white-colored product was precipitated, followed by filtration and washing with methanol, and finally dried in an open atmosphere.

Palladium(II) organophosphine complex (**1**) was prepared by reacting PdCl_2_ dissolved in methanol (30 mL) along with 4 drops of concentrated HCl with organophosphine dissolved in acetone (25 mL) in 1:2. The reaction mixture was reflexed and stirred for 3 h (Figure 1). The product was filtered, washed with methanol several times, and dried in a vacuum.

Heteroleptic or mixed ligand palladium(II) complexes (**2**–**4**) were prepared by the reaction of potassium salt of dithiocarbamate dissolved in dichloromethane (25 mL) with palladium phosphine complex dissolved in dichloromethane (30 mL), Figure 1. The reaction mixture was kept on reflux with vigorous stirring until the completion of the reaction. Thin-layer chromatography was used to monitor the development of the reaction. Upon the completion of the reaction, the solid by-product was filtered, the solvent was evaporated by a rotary evaporator, and the product was dissolved in a suitable mixture of solvents for recrystallization.

#### 3.2.1. Dichlorido(diphenyl–*n*–propylphosphine)palladium(II) (1)

An amount of 0.08 g (0.45 mmol) of PdCl_2_ was reacted with 0.20 mL (0.90 mmol) of diphenyl–*n*–propylphosphine. Yield: 79% (0.23 g). Melting point: 161–162 °C. Elemental analysis for complex **1** having a molecular formula C_30_H_34_Cl_2_P_2_Pd: % Calculated (found): H, 5.41 (5.39); C, 56.85 (56.71). **FT-IR** (cm^−1^): 2922, 2854 υ(C–H, aliphatic), 3052 υ(C–H, aromatic). **Raman** (cm^−1^): 298 υ(Pd–Cl), 221 υ(Pd–P). **^1^H NMR** (ppm): 0.96 (t, ^3^*J*: 7.2 Hz) C***H***_3_CH_2_CH_2_-, 1.49–1.55 (m) CH_3_C***H***_2_CH_2_-, 2.39–2.44 (t, ^3^*J*: 7.2) CH_3_CH_2_C***H***_2_-, 7.25–7.71 (m) Ar-***H***. **^13^C NMR** (ppm): 14.7 (***C***H_3_CH_2_CH_2_-), 23.0 (CH_3_***C***H_2_CH_2_-), 28.9 (C*H*_3_CH_2_***C***H_2_-), 128.3, 128.4, 134.5, 134.6 (Ar-***C***). **^31^P NMR** (ppm): 17.1.

#### 3.2.2. Chlorido–(dimethyldithiocarbamato–κ^2^S,Sʹ)(diphenyl-2-pyridylphosphine) palladium(II) (2)

A total of 0.05 g (0.41 mmol) of potassium salt of dimethyldithiocarbamate was reacted with 0.29 g (0.41 mmol) of Pd-phosphine complex. Yield: 78% (0.23 g). Melting point: 197–198 °C. Elemental analysis for complex **2** having a molecular formula C_20_H_20_ClN_2_PPdS_2_: % Calculated (Found) S, 12.21 (12.26); N, 5.33 (5.30); H, 3.84 (3.87); C, 45.72 (45.80). **FT-IR** (cm^−1^) analysis: 1096 υ(SCS), 1548 υC
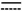
N, 2925, 2861, υ(C-H, aliphatic), 3056 υ(C-H, aromatic). **Raman** (cm^−1^): 380 υ(Pd–S), 300 υ(Pd–Cl), 218 υ(Pd–P). **^1^H NMR** (ppm): 2.80 (s), 3.58 (s) (C***H***_3_), 7.37–8.73 (m) (Ar-***H***). **^13^C NMR** (ppm): 35.5, 38.6 (***C***H_3_), 125.0, 128.9, 130.0, 130.8, 131.3, 134.7, 134.6, 134.9, 149.3, 157.6 (Ar–***C***) 207.5 (S***C***S). **^31^*P* NMR** (ppm): 27.5.

#### 3.2.3. Chlorido–(dimethyldithiocarbamato–κ^2^S,Sʹ)(diphenyl–2–ethoxyphenyl phosphine)palladium(II) (3)

A total of 0.03 g (0.25 mmol) of potassium salt of dimethyldithiocarbamate was reacted with 0.19 g (0.25 mmol) of Pd-phosphine complex. Yield: 85% (0.17 g). Melting point: 223–224 °C. The elemental analysis for complex **3** has a molecular formula. C_22_H_23_ClNOPPdS_2_: % Calculated (found) for: S, 11.57 (11.60); N, 2.53 (2.57); C, 47.66 (47.70). **FT-IR** (cm^−1^): 1097 υ(SCS), 1528 υ(C
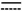
N), 2930, 2850 υ(C–H, aliphatic), 3052 υ(C–H, aromatic). **Raman** (cm^−1^): 390 υ(Pd–S), 301 υ(Pd–Cl), 215 υ(Pd–P). **^1^H NMR** (ppm): 3.14 (s), 3.26 (s) (C***H***_3_), 3.98 (q, –C***H***_2_), 1.33 (s, C***H***_3_), 6.90–7.76 (m, Ar–***H***). **^13^C NMR** (ppm): 14.3 (-CH_2_***C***H_3_), 65.2 (-***C***H_2_CH_3_), 120.1, 126.3, 128.2, 130.4, 131.7, 133.2, 164.0 (Ar-***C***), 208.2 (S***C***S). ^31^***P*** NMR (ppm): 20.3.

#### 3.2.4. Chlorido–(dimethyldithiocarbamato–κ^2^S,Sʹ)(diphenyl–*p*–tolylphosphine) palladium(II) (4)

A total of 0.04 g (0.33 mmol) of potassium salt of dimethyldithiocarbamate was reacted with 0.24 g (0.33 mmol) of Pd-phosphine complex. Yield: 77% (0.19 g). Melting point: 216–217 °C. Elemental analysis for complex **4** having a molecular formula C_22_H_23_ClNPPdS: % Calculated (found) for: S, 11.91 (11.87); N, 2.60 (2.64); H, 4.31 (4.40); C, 49.08 (49.00). **FT-IR** (cm^−1^) analysis: 1092 υ(SCS); 1530 υ(C
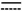
N) 2935, 2852, υ(C–H, aliphatic), 3040 υ(C–H, aromatic). **Raman** (cm^−1^): 395 υ(Pd–S), 308 υ(Pd–Cl), 210 υ(Pd–P). **^1^H NMR** (ppm): 3.09 (s), 3.27 (s) (C***H***_3_), 2.38 (s) (Ph–C***H***_3_), 7.43–7.92 (m) (Ar–***H***). **^13^C NMR** (ppm): 36.4, 39.1 (***C***H_3_), 128.3, 129.5, 134.0, 136.5, 136.8, 137.1, 137.6, 141.7 (Ar–***C***), 22.5 (Ph–***C***H_3_), 207.3 (S***C***S). **^31^*P* NMR** (ppm): 22.8.

### 3.3. Biological Activity Assays

#### 3.3.1. Antibacterial Activity Assay

The antimicrobial potential of the prepared compounds was evaluated using the agar-well diffusion method against five bacterial strains: two Gram-negative and three Gram-positive strains: *E. coli*, *K. pneumoniae*, *S. epidermidis*, *S. aureus*, and *B. subtilis* [39]. Using a sterile cotton swab, about 10^4^–10^6^ colony-forming units (CFU)/mL of bacterial inoculums were dispersed on top of nutritional agar. The tested compounds having a concentration of 2 mg/mL in DMSO were transferred to each well, and then the plates were incubated for 20 h at 37 °C, and finally the inhibition zone (in mm) was measured to determine the antibacterial activity. Streptomycin was used as the standard drug and DMSO as a positive control for the determination of growth inhibition, and the experiment was performed three times [39].

#### 3.3.2. Antitumor Activity

Using the technique described in the referred articles, the anticancer activity of the synthesized compounds was assessed against DU145 human prostate cancer (HTB-81) cells [40,41,42]. Cisplatin was used as a reference drug. Compounds with a concentration of 50 mmol were dissolved in DMSO and diluted into nine consecutive concentrations, with the final concentration of DMSO on the cells not exceeding 0.05 percent. In 96-well flat-bottom microtiter plates, DU145 prostate cancer cells were planted at a density of 5000 cells per well for the growth inhibition test. Following a 24-h incubation period, cells were treated for four days with varying doses of each treatment. The remaining viable cells were fixed with 50% cold trichloroacetic acid for 60 min at 4 °C, stained with 0.4% sulforhodamine B (SRB) for four hours at 25 °C, washed with 1% acetic acid, and allowed to dry overnight. After dissolving the colored residue in 10 mM Tris base (pH = 10) and at 490 nm using an ELx808 microplate reader, the optical density was recorded. Graph Pad Prism was applied for data analysis, and the sigmoidal dose response curve was used to compute the IC_50_. The test for growth inhibition was repeated three times [40,41,42].

### 3.4. In Silico Studies

The attrition rate for clinical trials used to develop new drugs has increased to 90% during the last ten years. Over a five-year period, on average, 26.8 small molecules were approved as FDA drugs. Only 12 innovative small-molecule medicines were approved by the FDA in 2016—the fewest such approvals over the previous fifty years [43,44]. Pharmaceutical firms invest millions of dollars to push a new treatment through clinical trials; therefore, failure in the latter stages of drug development often results in large financial losses [45]. The major reasons why drug candidates fail in clinical trials are undesirable pharmacokinetic characteristics and unacceptable toxicity [46]. Therefore, it is crucial for science to select candidates with the right balance of potency along with absorption, distribution, metabolism, excretion, and toxicity (ADMET). Parameters for ADMET and drug-like properties are given in Table 5.

### 3.5. Molecular Docking Analysis

The binding mode and affinity of a small molecule or ligand with a macromolecule, such as a protein or DNA, can be predicted using molecular docking. In the case of drug discovery, molecular docking is used to identify potential drug candidates that can bind to a target protein with high affinity and specificity. Here we have treated Palladium (II) complexes against three bacterial strains: *K. pneumoniae*, *S. aureus*, and *E. coli,* as well as trial receptor DR5, for which the PDB files were obtained from the RCSB PDB homepage.

MOE-Dock software version 2015 was used to perform docking studies so as to identify the binding interactions of the screened compounds in the active site of three antibacterial targets, such as PDB_ID: 4EXS from *Klebsiella pneumonia* [47], PDB_ID: 5ZH8 from *Staphylococcus aureus* [48,49], and PDB_ID: 6G9S from *Escherichia coli* [50], as well as death receptor (DR5) PDB_ID: 1DU3. The compounds were built in MOE, and energy was minimized by using the MOE’s default settings parameters, Placement: Triangle Matcher and Rescoring: London dG, for the docking study [51]. Two conformations were generated for each ligand. For further molecular interaction analysis, the highest-ranked conformation of each compound with the lowest binding energy score was used. The interactions between receptor–solvent and ligand–solvent were excluded during the generation of the 2D interaction diagram.

## 4. Conclusions

A total of four Pd(II) complexes, including one homoleptic and three heteroleptic, have been prepared quantitatively. The single crystal analysis of complex **1** shows that the geometry around the palladium atom is square planar with slight distortion. The synthesized compounds have shown significant antibacterial activity against the selected targets. Compound **2** has shown the maximum antitumor activity against DU145 human prostate carcinoma cells. The lower binding energy or higher inhibition values indicate a stronger binding affinity of complex **3** with the target proteins. The molecular docking study of the evaluated compounds with DR5 (1BU3) shows the highest binding affinity for compound **2** (−7.5148 kcal/mol) and then for compound **3** (−7.0343 kcal/mol). In silico studies have been performed by the SwissADME webserver, and the compounds generally have shown one or two violations of Lipinski’s rule of five.

## Data Availability

Data will be available on request to corresponding authors.

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
