# Peer review of "Synthesis, Physicochemical Characterization, Biological Evaluation, In Silico and Molecular Docking Studies of Pd(II) Complexes with P, S-Donor Ligands"

_pharmaceuticals, 2023, doi:10.3390/ph16060806_

Round 1

Reviewer 1 Report

-In the Abstract part avoid the abbreviations like “E. coli, K. pneumonia and S. aureus”.

-In the Abstract part add in the method subsection, the name of the used anticlerical assay.

-Add briefly to the abstract part antibacterial results.

-Add statistical analysis subsection to the material and method section.

-Add statistical analysis to all obtained results of your study.

At the end of abstract part conclusions and future recommendations.

Nothing in the keyword mentioned about antibacterial activity.

- The whole manuscript needs language and editing corrections 

- I recommend authors to compare their antibacterial results with antibiotics as positive control.

-throughout the whole manuscript and other Latin words must be in Italic manner.

The whole manuscript needs language and editing corrections by a native speaker

Author Response

Dear Reviewer

Thanks a lot for your valuable suggestions and corrections. we have tried our best to consider each and every point you have mentioned.

-In the Abstract part avoid the abbreviations like “E. coli, K. pneumonia and S. aureus”.

Author Response: Full names are now used instead of abbreviations: Escherichia coli, Klebsiella pneumonia and Staphylococcus aureus

-In the Abstract part add in the method subsection, the name of the used anticlerical assay.

Author Response: Agar well diffusion method was used for antibacterial and using the sulforhodamine B (SRB) method was used for anticancer and are now added in the abstract as suggested.

-Add briefly to the abstract part antibacterial results.

Author Response: Done as suggested.

-Add statistical analysis subsection to the material and method section.

Author Response: Graph Pad Prism was applied for statistical analysis which is now mentioned in the material and method section.

-Add statistical analysis to all obtained results of your study.

Author Response: Added now as suggested.

At the end of abstract part conclusions and future recommendations.

Author Response: Recommendation added.

Nothing in the keyword mentioned about antibacterial activity.

Author Response: It is now added in the keywords.

- The whole manuscript needs language and editing corrections 

Author Response:  The whole manuscript is now thoroughly checked by an English Expert to omit the mistakes.

- I recommend authors to compare their antibacterial results with antibiotics as positive control.

Author Response: It can be seen in antibacterial Table 4, where Streptomycin is mentioned as positive reference drug.

-throughout the whole manuscript and other Latin words must be in Italic manner

Author Response: Done as suggested.

Reviewer 2 Report

This is a well-designed and performed study. They used several experimental approaches: chemical synthesis, physicochemical characterization, antibacterial activity assays, antitumor activity and in silico and molecular docking studies. The results obtained are very relevant.

Major point:

A conclusion must be added at the end both of the manuscript and abstract and the abstract.

Minor:

Line 296: “…of 2>3>4>1.” Should be “…of 2>3>1>4.”

Author Response

Dear Reviewer, We are highly thankful for your valuable time and mentioned some critical points and suggestions. All the corrections made are yellow highlighted in the paper.

Comments. I suggest that authors provide a hypothesis. The justification for the use of dithiocarbamates for its incorporation to complexes with palladium is weak since use of references is missing.

Author Response: A paragraph including references (23-26) is added in the introduction part.

Please: Uniformize the use of words “cis-platin” or “cisplatin”.

Author Response: Ok suggestion incorporated. cisplatin is used.

Line 71. It is necessary to indicate the reference number of Khan et al.

Author Response: Ref. # 17-19 are now mentioned.

Line 113. Correct “dithiocarbamte” word.

Author Response: Corrected.

Line 304. Correct “positve” word.

Author Response: Corrected.

Line 182. Eliminate the abbreviation in the following phrase “The tested compounds having conc. 2 mg/mL”

Author Response: Done 

Round 2

Reviewer 1 Report

The authors conducted all the required corrections and I have not any more corrections